


**Observations and scaling of tidal mass transport across**
**the lower Ganges-Brahmaputra delta plain:**
**implications for delta management and sustainability**
Richard Hale[1], Rachel Bain[2], Steven Goodbred Jr.[2], Jim Best[3]
[1]Dept. of Ocean, Earth, and Atmos. Sci., Old Dominion University, Norfolk, VA, USA
[2]Earth and Environmental Sciences Dept., Vanderbilt University, Nashville, TN USA
[3]Departments of Geology, Geography & GIS, Mechanical Science and Engineering and Ven
Te Chow Hydrosystems Laboratory, University of Illinois, Champagne, IL USA
**Abstract**
The landscape of southwest Bangladesh, a region constructed primarily by fluvial
processes associated with the Ganges and Brahmaputra Rivers, is now maintained almost
exclusively by tidal processes as the fluvial system has migrated to the east through the
Holocene. In natural areas such as the Sundarbans National Forest, year-round spring-tide
inundation delivers sufficient sediment for vertical accretion to keep pace with relative sea-
level rise. However, recent human modification of the landscape in the form of
embankment construction has terminated this pathway of sediment delivery for much of
the region, resulting in a startling elevation imbalance, with inhabited areas often sitting >1
m below mean high water. Restoring this landscape, or preventing land loss in the natural
system, requires an understanding of how rates of water and sediment flux vary across
time scales ranging from hours to months. In this study, we combine time-series
observations of water level, salinity, and suspended sediment concentration, with ship-
based measurements of large tidal channel hydrodynamics and sediment transport. To
capture the greatest possible range of variability, cross-channel transects designed to
encompass a 12.4-h tidal cycle were performed in both dry and wet seasons, during spring
and neap tides.
Regional suspended sediment concentration begins to increase in August, coincident with a
decrease in local salinity, indicating the arrival of the sediment-laden, freshwater plume of
the combined Ganges-Brahmaputra-Meghna rivers.  We observe profound seasonality in
sediment transport, despite somewhat modest seasonal variability in the magnitude of
water discharge, indicating the importance of this seasonal sediment delivery. On tidal
time-scales, spring tides transport an order of magnitude more sediment than neap tides in
both the wet and dry seasons. In aggregate, sediment transport is flood-oriented,  likely a
result of tidal pumping. Finally, we note that rates of sediment and water discharge in the
tidal channels are of the same scale as the annually averaged values for the Ganges or
Brahmaputra rivers. These observations provide context for examining the relative
importance of fluvial and tidal processes in what has been defined as the quintessential
tidal delta in the classification scheme of Galloway. These data also inform critical
questions regarding the timing and magnitude of sediment delivery to the region, which
are especially important in predicting, and preparing for, future change under changing
environmental conditions.





## 1 – Introduction

The world's great deltas are currently threatened by a variety of factors, including global sea level rise (Overeem and Syvitski, 2009), overpopulation (Ericson et al., 2006), changes in sediment supply (Syvitski 2003; Syvitski and Milliman, 2007; Anthony et al., 2015; Darby et al., 2016), and other human-related activities such as water diversions, flood control structures, and groundwater and hydrocarbon extraction (Syvitski et al., 2009). The Ganges-Brahmaputra-Meghna (GBM) delta is one of the most heavily populated regions that is undergoing locally accelerated sea-level rise (~0.5 cm/y; Higgins et al., 2014) due to a combination of natural and anthropogenic factors including eustatic sea-level change, tectonic subsidence, fine-grained sediment compaction, and groundwater extraction (Overeem and Syvitski, 2009; Syvitski, 2008; Steckler et al., 2010). In addition, when factors like tidal amplification due to anthropogenic reworking of the distributary channel network are considered, the relative rate of sea-level rise can exceed 1.6 cm/yr (Pethick and Orford, 2013). Furthermore, the future viability of the delta is threatened by the proposed construction of dams and water diversions associated with the India River Linking Project, which, if completed as proposed, could drastically reduce sediment delivery to Bangladesh (Higgins et al., 2018).

Restoration of land-surface elevation in many populated areas in the GBM delta is already necessary due to the relative loss in elevation that has occurred since the widespread construction of embankments during the 1960s to 1980s. Both planned (tidal river management) and unplanned (embankment failures) flooding of local polders (the embanked islands) has demonstrated the capacity of the natural system for effective sediment transport and deposition, with decimeters of annual accretion observed during recent breach events (Khadim et al., 2013; Auerbach et al., 2015; Kamal et al., 2017; Darby et al., 2018). One of the most important strategies that has been forwarded to reduce the threat of unintended inundations in SW Bangladesh is a plan for polder management (Brammer, 2014). However, many questions concerning potential management strategies remain, not the least of which are an accurate quantification of total available sediment mass and an understanding of the tidal processes involved in its transport and deposition. Toward these goals, the present study provides observation-based calculations of water and sediment transport through a major tidal channel in the delta across spring-neap tidal cycles and seasonal time scales, with the goal of identifying the timing and magnitude of mass sediment exchange between the different tidal channels. These results are then considered in the context of prior research concerning sediment accumulation and rates of channel infilling to better understand the role of tidal mass transport within the lower GBM delta plain.

## 2 – Background

Much of the low-lying coastal region of SW Bangladesh is under threat of long-term inundation (Auerbach et al., 2015; Brown and Nicholls, 2015). The risk is particularly acute for islands that were embanked ("poldered") in the 1960s and 1970s as part of a program designed to increase the area of arable land through the prevention of tidal inundation in agricultural areas (Islam, 2006; Nowreen et al., 2014). Approximately 5000 km of polder



embankments were built by hand, generating 9000 km² of new farmland, but also
eliminating the semi-diurnal exchange of water and sediment between the tidal channels
and tidal platform (Islam, 2006; Nowreen et al., 2014). As a result, sediment resupply
pathways have been effectively terminated and the former floodplain surface in these
regions now lies 1.0-1.5 m below mean high water due to a combination of sediment
starvation, enhanced sediment compaction, and tidal-range amplification (Auerbach et al.,
2015; Pethick and Orford, 2013).
In contrast to the poldered landscape, the adjacent mangrove system of the Sundarbans
National Forest (SNF) is primarily inundated during spring high tides, and its
sedimentation and vegetation are keeping pace with sea-level rise (Rogers et al., 2013;
Auerbach et al., 2015;). Protecting the SNF is of critical importance, as coastal wetlands and
mangroves provide irreplaceable ecosystem services including storm-surge buffering
(Uddin et al., 2013; Marois and Mitsch, 2015; Hossain et al., 2016; Sakib et al., 2015),
serving as effective carbon traps (Mcleod et al., 2011;  Alongi, 2012; Pendleton et al., 2012)
and perhaps even helping to combat the impacts of ocean acidification (Yan, 2016).
For the GBM delta, a unit-scale analysis of mass balance (Rogers et al., 2013) suggests that
the annual sediment load of the GBM river system (~1.1 Gt/y) is sufficient to aggrade the
entire delta system at rates ≥ 0.5 cm/yr, and thus provides potential to keep pace with
moderately high rates of sea-level rise. Such aggradation, of course, requires effective
dispersal of riverine sediment to disparate regions of the delta. Recent research suggests a
close coupling of discharge at the river mouth to sediment deposition in the remote delta
plain by way of tidal exchange (Allison and Kepple, 2001; Rogers et al., 2013; Auerbach et
al., 2015; Wilson et al., 2017). Such tidally supported sedimentation yields mean accretion
rates of ~1 cm/ yr, with local observations regularly reaching 3-5 cm/yr, which together
indicate robust sediment delivery to the Sundarbans and SW coastal region (Rogers et al.,
2013).  Thus, as the principal conduit for sediment that can maintain the elevation of this
region, an understanding and quantification of the tidal water and sediment exchange is
essential to foresee future impacts of accelerated sea-level rise and the potential for
mitigation.

**3 – Methods**

**3.1 – Study Area**
Our research concerns a network of tidal channels located approximately 80 km from the
coast along the Pussur River system, itself one of five similarly sized tidal distributary
networks (Fig. 1). Tidal exchange extends >120 km inland of the coast along the Pussur
River, with one branch ultimately connecting to the Gorai River, a distributary of the
mainstem Ganges River (Fig. 1). The tidal range along the Pussur River approaches its
maximum in the study area at 4-5 m for the spring tidal range, as compared with 3-3.5 m
on the coast at Hiron Point.  The area is also societally relevant, lying at the transition from
the pristine Sundarbans to the embanked polders, and near the formerly active shipping
port of Mongla and cyclone- and flood- impacted island of Polder 32 (labelled P32 on Fig. 1;
Auerbach et al., 2015).



Within this area, our observations were collected in the primary tidal channel of the Shibsa
River and two of its major bifurcations that connect with the Pussur channel, the Dhaki
River and Bhadra River (Fig. 1).  The Shibsa River is the largest of these channels, with local
widths of 1-2 km, compared to 0.25-0.8 km and 0.15-0.3 km, for the Dhaki and Bhadra
Rivers, respectively. At its eastern extent, the Dhaki River connects to the Pussur River,
serving as the first major cross-channel to link the Shibsa and Pussur River channels after
they bifurcate ~60 km to the south (Fig. 1). At its upstream extent, the Pussur tidal channel
connects with the downstream mouth of the Gorai River, which delivers a water discharge
of ~3000 m³/s during the monsoon season to ~0 m³/s during the dry season (Winterwerp
and Giardino, 2012). Salinity in the study area ranges from 0-1 PSU during the monsoon, to
20-30 PSU during the dry season (Shaha and Cho, 2016; Ayers et al., 2018). This seasonal
variation in salinity is controlled by local runoff, the freshwater discharge from the Gorai
River, and to a much larger extent, the magnitude of the regional plume of the GBM rivers
(Rogers et al., 2013).
**3.2. – Hydrodynamic Observations**
To establish tidal stage and capture surface-water elevations during the hydrodynamic
surveys, pressure sensors were deployed at multiple locations across the study area (Fig.
1). All sensors were deployed as close to low water as possible and recorded at 5- or 10-
minute intervals.  Periods of subaerial sensor exposure (of up to 150 minutes at low tide)
were interpolated using a robust ordinary least-squares method provided by Grinsted
(2008). The agreement between measurement and prediction was generally good, with
predicted range being 0.98 of the measured range for a given time period, thus suggesting
that the interpolated data are both reasonable and conservative. The values reported
herein are of the interpolated values. Tidal range, water temperature, and conductivity
have also been monitored continuously since 2014 at the Sutarkhali station (Fig. 1B), with
an optical backscatter sensor (OBS) to measure suspended sediment concentration (SSC)
added in late March 2015. While the sediment concentrations recorded by this near-bed
instrument are not directly comparable to the depth-averaged measurements made during
the present cross-channel surveys, we herein use these data to extend our understanding
of system behavior between the dry and monsoon seasons. For broader context, data from
the sensors deployed at the Sutarkhali station are also compared to monthly averaged
water discharge for the Ganges and Brahmaputra rivers for the period 1980-2000, based
on data from the Bangladesh Water Development Board, and Ganges River sediment
discharge data digitized from Lupker et al. (2011).
To quantify water and sediment flux in this area of the tidal transport system, cross-
channel hydrodynamic surveys were conducted during spring and neap tidal conditions at
two transects on the Shibsa River during the dry (March 2015) and wet
(August/September 2015) seasons. An additional wet season transect was also conducted
during moderate tides on the Pussur River. On the Shibsa River, the southern transect was
located south of the poldered landscape and entirely within the confines of the SNF (Fig.1).
The northern transect was located ~12 km upstream in the poldered region, just south of
the Dhaki-Shibsa confluence and adjacent to Polder 32 to the east and Polder 10-12 to the
west (Fig. 1B). Two secondary channels are present between these transect locations that



divert water onto the Sundarbans tidal platform and associated creek network. Dry season
surveys at both the southern and northern transects took place during peak neap (15-16
Mar) and spring (21-22 Mar) tides.  During the ensuing monsoon season, spring tides were
measured on August 30-31 (southern transect) and September 2 (northern transect),
followed by neap tides on September 7 and 8 (northern and southern transects,
respectively). Surveys lasted for 11-13 hours as conditions allowed, encompassing
approximately one-half of a tidal cycle (i.e., one high and one low tide). Because this system
is largely semi-diurnal with a minimal mixed component, we are confident that this time
interval was long enough to describe the system dynamics accurately.
The surveys were conducted using Sontek M9 multi-frequency ADCP's to collect flow-
perpendicular observations of current velocity and direction. Data were collected at 1 Hz,
using both 1.0 and 3.0 MHz transducers, resulting in vertical bins ranging in height from
0.1-1.0 m. From these values, total discharge was calculated by integrating velocity over
space and time. River conventions are used for presenting velocity and discharge data,
where positive values refer to the ebb or downstream direction and negative values for the
flood or upstream transport. A typical survey day included 50-60 individual river crossings
at the transect location, measuring cumulative discharge in both directions across the
channel. Because surveys could only be conducted during daylight hours and as weather
conditions allowed, discharge is interpolated to complete a 12.4-hour tidal cycle, which is
the average tidal cycle duration in the area (range: 11.9-13.1 h).  By assuming that the
change in tidal prism is negligible between consecutive tides, as suggested by the similarity
in tidal elevations (Fig. 2), we can tile measurements in 12.4 h increments and interpolate
using a cubic spline.  Working conditions were particularly challenging during the monsoon
season, resulting in especially short-duration survey days. In the absence of measured
discharge, we use a mass balance approach to constrain the magnitude of the missing tidal
prism data. For the monsoon-season spring tides, we treat the region between the southern
and northern transects and the southern Bhadra River as a closed system with no long-
term (>1 semidiurnal period) water storage. Using measured Bhadra River discharge
values and assuming a negligible to slightly southerly-directed net flux through the
adjacent Sundarbans, allows us to determine the likely range of values for the unmeasured
ebb prism at the southern transect. For the monsoon-season neap tides, we consider the
larger region bounded by the southern transect to the southwest, the Pussur River below
the Dhaki River confluence to the southwest, and the Bangladesh Water Development
Board gauging station at the Gorai Railway Bridge ~275 river km to the north. Balancing
the measured net flux through the Pussur River and the recorded upstream discharge of
the Gorai River of 3000 m³/s  with the measured ebb prism at the southern transect allows
us to estimate the missing southern transect flood prism. We then repeated this spring tide
procedure to estimate the unmeasured neap flood prism at the northern transect.
**3.3 – Sediment Observations**
In addition to water discharge, observations of SSC along the transect lines were made
using a combination of filtered water samples and optical-backscatter (OBS)
measurements. While the exact sampling method varied depending on available
instrumentation and river conditions, the general approach involved collecting OBS



profiles to the maximum possible depth (<10 m), at either two (northern transect) or three
(southern transect) locations along the channel edges and centerline (Fig. 1). OBS
measurements were supplemented by simultaneous water samples (100-200 ml) collected
from various depths using a Niskin sampler. Water samples were filtered using pre-
weighed 0.4-µm nitrocellulose filters and washed with freshwater to remove salts. The
filters were then dried overnight and re-weighed to determine the volume-concentration of
sediment. The OBS measurements were calibrated by comparing the voltage response
observed in the field with the measured concentrations from the same time and location, in
a method modified from Ogston and Sternberg (1999). Correlation between filtered
samples and instrument voltage was strong, with an average r-squared value of 0.83±0.1.
In order to calculate the total sediment fluxes, the vertically and horizontally distributed
SSC observations collected for each channel cross-section were averaged to produce a
series of temporally discrete SSC values over the course of one tidal cycle. This spatial
averaging appears suitable because the variance was considerably smaller than the
temporal variability associated with tidal discharge and strong seasonal contrasts. Using
wet season data as an example, the average standard deviation of SSC through time at one
sample location was 0.2 g/L, while the average standard deviation of SSC between stations
at any given time was 0.13 g/L. When conditions did not allow samples to be collected at
depths below the water surface, a scaling factor of 1.25 was applied to account for the
higher sub-surface SSC, which we determined by the relationship between depth-averaged
concentrations and surface concentrations from the other available data. Similarly,
measurements from 15 March (dry-season neap tide) were only collected at depths of 5
and 15 m and were thus scaled by a factor of 0.81 to be comparable to other measurements
that included surface SSC values. An important caveat for all SSC measurements is that we
are referring primarily to suspended load, largely ignoring the bedload component. While
this is presumably a measurable component of the total sediment transport, bedload is
likely unable to exit the tidal channels during platform irrigation, and as such is not
considered an important source of sediment for land construction. As with our
measurements of water discharge, SSC values were calculated over an entire tidal cycle by
tiling observations in 12.4-hour segments and interpolating using a cubic spline. From
these values, the integrated product of water discharge and SSC yields net sediment flux,
which we compute using the time series for each component as calculated using the
aforementioned methods.
**4 – Results**
**4.1 – Long-term Pressure and OBS:**
At our long-term station deployed in a secondary tidal channel (Fig. 1), recorded water-
level variations show tidal-period excursions with a range of 1.8 to 4.8 m over the 12
months of observation (Fig. 2).  This variance is, of course, driven primarily by the
fortnightly spring-neap tidal cycle, but there is also a seasonal variability showing the
monsoon period to have a reduced tidal range as compared with the dry season.  In this
case, the neap tidal range is ~10% less during the monsoon season, and the spring tidal

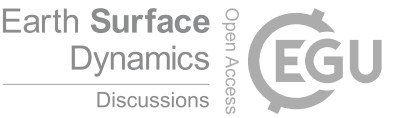

range is as much ~20% less, accounting for a nearly 1 m difference (3.9 m vs. 4.8 m). This
reduced range in the monsoon season, however, is not manifested in the elevation of high-
tide water levels, which remained largely consistent between seasons. Rather, the
difference is caused by higher water levels during low tide (Fig. 2), which has the effect of
truncating the tidal range and yielding an overall higher mean water level. These higher
low-water levels associated with the monsoon suggest that they are tied to regional
freshwater drainage and discharge. In addition, another contributing factor could be the
seasonally reversing wind stresses, but such set-up should enhance high water levels as
well, suggesting that they are not the primary cause.  Although further research on this
topic is needed, these distinctions are important herein for understanding the behavior of
the tidal delta plain, as landscape elevations in this region are closely tied to mean high-tide
water levels, and not mean sea level (Auerbach et al., 2015). Thus, as first demonstrated by
Pethick and Orford (2013), the monthly mean tide-gauge data often used to track seasonal
to interannual variations in water level may have relatively little bearing on the tidal
inundation period and sedimentation rates that control tidal platform elevation (Rogers et
al. 2013).
The arrival of fully freshwater (wet-season) conditions occurs in July, following the peak in
Brahmaputra River water discharge, and roughly coincident with peak Ganges River water
discharge (Fig. 3). Coupled with our long-term pressure gauge, the OBS sensor recorded
relatively constant, but low, mean SSC from the late dry season into the early monsoon
period (late March through July), with weak but noticeable spring-neap variability ranging
from ~0.01 g/L to 0.20 g/L (Fig. 2). However, moving into peak monsoon season, SSC
increases markedly from early August through September, concurrent with the Ganges
River sediment discharge peak (Figs. 2, 3).  Individual measurements regularly exceeded
0.50 g/L during this time, with maxima >2.5 g/L (Fig. 2). SSC variability around the semi-
diurnal tide and spring-neap cycles was greatly enhanced compared with that during the
dry season, with SSC values during spring tidal cycles exceeding those observed during
neap conditions by a factor of 2-10. By the end of observations in October 2015, SSC began
to drop to levels similar to those observed in mid-August (0.01-1.0 g/L; Fig. 2), but on
average remained well above those of the dry season. For comparison, the mean annual
SSC of the mainstem Ganges-Brahmaputra river is ~1 g/L, and depth-averaged values in
the main estuary mouth and on the inner shelf commonly range 2-5 g/L during high river
discharge (Barua et al., 1994; Ali et al., 2013). In total, SSC values well in excess of 1 g/L are
regularly observed during the wet season from the mainstem river to the inner shelf and
into the tidal channels of the lower delta plain. These results support previous evidence for
the strong coupling of seasonal river discharge with penecontemporaneous sedimentation
in the remote tidal delta plain (Rogers et al., 2013).
**4.2 – Hydrography – Water Discharge:**
Dry season tidal range on the Shibsa River, as measured at Nalian near the northern
transect (Fig. 1B), varied from 2.3 m during the neap minima to 5.6 m during spring
maxima (Fig. 2). The tidal period was slightly longer during neap tides than spring tides
(12.9 h vs. 12.3 h), and the mixed component of the semi-diurnal tide was more
pronounced, with consecutive tidal ranges varying by as much as 0.55 m during neap tides,



versus 0.23 m during spring tides (Fig. 2).  During the monsoon fieldwork, the tidal range
was 2.4 and 4.2 m for the neap and spring tides respectively. As with the dry season, total
tidal period during neap tides was slightly longer than spring tides (12.8 h vs. 12.0 h). The
mixed semi-diurnal variability was again greater during neap tides as well, which varied by
as much as 0.25 m, while spring tide variability was typically <0.10 m (Fig. 2).
In order to calculate the dry-season tidal prism (i.e., integrated ebb and flood discharges),
our observations captured both peak flood and ebb discharges, with interpolation being
used over the remaining <5-15% of the tidal cycle (Fig. 4). During the monsoon season,
challenging field conditions resulted in several surveys capturing only a partial tidal cycle
(~8-9 hr survey; Fig. 4). Only during northern transect spring tides were conditions
favorable for collecting observations of similar duration to the dry season (~11 hr survey;
Fig. 4). Within these limits, however, we have used conservative interpolation to generate
error-bound estimates of total water discharge, the resulting patterns of which provide
robust observations concerning system behavior (see Section 2; Fig. 4).
The average tidal prism magnitudes for the northern and southern transects are $2.1\pm0.7 \times$
$10^8$ $m^3$ and $3.4\pm1.4 \times 10^8$ $m^3$, respectively. Included in these averages the absolute values of
flood and ebb tidal prisms measured on spring and neap tides during both wet and dry
seasons (Table 1).  Thus, the tidal prism at the northern transect averages only ~60±10%
that of the southern transect regardless of season, even though they are located just 10 km
apart. Most of this difference in discharge (*c.* 80-100%) can be balanced by water storage
between the two locations, where the product of tidal range and area between transects is
*c.* $6.7 \times 10^7$ $m^3$.  Considering differences in seasonal discharge, results show that the neap
ebb prism is ~30% greater during the monsoon at both transects, despite having a smaller
tidal range compared with the dry season survey. This difference of $4$-$6 \times 10^7$ $m^3$ equates to
an excess ebb discharge of 1800-2800 $m^3$/s, which is about 45-70% of the mean monsoon
discharge of the upstream Gorai river.  We thus take the greater wet-season ebb prism to
simply reflect the addition of local freshwater discharge from the Gorai River (Table 1; Fig.
352    1).
Strictly speaking, defining a tidal regime as either ebb- or flood- dominant refers to the
water velocity rather than discharge (Pethick, 1980; Brown and Davies, 2010). In the
present study, however, we are interested in the net movement of water and sediment and
thus refer to a particular discharge regime as either ebb or flood "dominated" or "oriented"
based on the net tidal prism (i.e., the difference between ebb and flood discharge). With
this in mind, our surveys suggest that the system varies between ebb and flood orientation
across both tidal phase and season (Table 1).  For example, both transects during the dry,
spring and wet, neap surveys show the average ebb-tidal prism to be 26±16% larger than
the flood limb.  In contrast, the other two survey periods (dry, neap and wet, spring)
yielded balanced to slightly flood dominated tidal prisms (9±8%).  In summary, although
our results on water balance are insufficient for a full understanding of the patterns, a key
finding is that the ebb and flood tidal prisms rarely balance at this location. These tidal-
prism asymmetries appear to be a salient characteristic of the complex, interconnected
channel network of the GBMD tidal delta plain. Thus, even our limited observations require



a lateral (east-west) exchange of water between the Shibsa and parallel Pussur channels
(Fig. 1), which we presume to be driven by locally non-uniform tidal phasing within the
channel network. Given these emergent circulation patterns, it is clear that individual
channels do not operate as closed systems and exhibit local, non-uniform mass exchange,
providing a first indication of how morphologic evolution of the tidal delta plain occurs.
Although the relative dominance between the ebb and flood tidal prisms covaries
persistently (as described above), the mean and instantaneous water discharge ($m^3$/s) is
almost always flood-dominant (Fig. 5). This circumstance arises from the significant phase
shift that occurs as the tide wave propagates up channel, resulting in a shorter flood period
and thus higher peak discharge. From our measures of instantaneous discharge across
seasons and tidal conditions, we calculate mean ebb and flood discharges ($m^3$/s) for each
transect (Fig. 5).  Mean discharge for the northern transect is ~9100 $m^3$/s on the flood and
8600 $m^3$/s on the ebb, and for the southern transect, mean flood and ebb discharges are
~14,600 and 14,200 $m^3$/s, respectively.  From these results, we observe that mean
discharge at the northern transect is again ~61±1% that of the southern transect, as also
recognized for the tidal prism.  Another notable result is that the mean flood discharge
($m^3$/s) is 3-6% greater than on the ebb tide, despite the tidal prism generally being ebb
dominant.  This disparity is a function of the shallow-water distortion of the M2 tide, which
produces an asymmetrical waveform with a steeper rising limb than falling limb, and a
corresponding reduction in the duration of the flooding tide by ~60-90 minutes.
**4.3 – Hydrography – Sediment Transport:**
Suspended sediment measurements collected during the hydrographic surveys show
similar patterns to those of our long-term OBS station. Wet season sediment concentrations
were generally 30-50% higher than during the dry season (Fig. 4). Much greater
differences in SSC were observed, however, between neap and spring tidal conditions, with
the latter concentrations being typically ~3 fold greater (0.3-1.5 g/L vs 0.1-0.5 g/L).  These
sediment concentrations, coupled with the water discharge observations, were then
extrapolated over the tidal cycle to generate estimates of the rates and magnitude of
sediment transport (Table 1). Results show that integrated sediment transport over a tidal
limb varied by more than an order of magnitude at both transects. Minima of $0.16 \times 10^8$ kg
(north) and $0.2 \times 10^8$ kg (south) were observed during the neap, dry-season ebb tide, and
maxima of $3.3 \times 10^8$ kg (north) and $3.9 \times 10^8$ kg (south) occurred on the spring, monsoon
flood tides.  These values equate to mean rates of sediment transport that range from ~700
kg/s during neap, dry season conditions to ~17,000 kg/s during monsoon-season spring
tides. Comparing the ebb and flood limbs of our surveys, the mean sediment discharge for
the ebb tide is 5800 kg/s compared to 7800 kg/s for the flood tide, demonstrating an
overall flood dominance in sediment transport.
These patterns are further supported by the net sediment transport values (i.e., ebb – flood;
Table 1). For a given tidal cycle, net sediment transport was typically $10^6$-$10^7$ kg, with
magnitude varying largely with tidal phase, where spring tides generate 1.5 to 3 times




greater net transport than during neap tides (Table 1). Seasonally, net sediment transport
rates were ~30% greater during the wet season, similar to our observations of suspended
sediment concentration.  Finally, a comparison of net sediment transport with
corresponding net water discharge shows the two to covary, as expected, with greater net
water discharge resulting in greater net sediment transport (Fig. 6). However, an important
attribute of this relationship reveals a significant bias toward flood-dominant sediment
transport. Data show that even neutral to weakly ebb dominant water discharge yields net
transport in the flood direction (Fig. 6).  As noted for water discharge (m$^3$/s), this disparity
is a function of the non-negligible tidal components beyond M2 that result in a shortened
flood limb and extended ebb period (Fig. 2; Table 1). Together, mean sediment discharge
and net sediment transport patterns thus indicate an overall flood-oriented asymmetry and
net onshore transport of sediment.
**5 - Discussion**
**5.1 – Relative importance of tides and river**
The GBM tidal delta plain comprises a complex channel network that has been little studied
and will require substantial investigation to be understood well. Nevertheless, results of
the current study allow for numerous observations on the scaling and magnitude of tidal
mass transport within this region, establishing a baseline for the role that tides play in
defining the delta system, particularly in the southwest region away from direct fluvial
inputs. To begin, we take an average of the flood and ebb tidal prisms measured at the two
sites on the Shibsa River over both spring and neap tidal phases during wet and dry
seasons, and extrapolate the mean tidal prism over one year. In other words, an average of
$2.7 \times 10^8$ m$^3$ water passes through this region on each of the ~705 tides per year. This basic
estimation accounts for an average of ~$2 \times 10^{11}$ m$^3$ of water annually conveyed through
our survey locations, 80 km inland of the coast.  Furthermore, this mass exchange is
principally tidal water, as the 50-75% of annual Gorai River discharge captured by the
Shibsa River (i.e., ~$0.2 \times 10^{11}$ m$^3$) accounts for only 10% of the total water exchange
observed for that channel.
The significance of these observations from the upstream Shibsa River tidal channel
become more apparent when compared with the mainstem GBM rivers. In this case, the ~2
$\times 10^{11}$ m$^3$ of water conveyed annually through the upper Shibsa River is nearly 20% of the
~$11 \times 10^{11}$ m$^3$ of annual discharge from the entire GBM watershed (Lupker et al., 2011; Fig.
3).  This is an impressive exchange of mass through the upper reaches of a single tidal
channel along the GBM tidal delta plain.  For context, the Shibsa River comprises
approximately half (by planform area) of the Pussur River tidal system (Fig. 1), itself just
one of five major tidal drainages along the GMB tidal delta plain (Fig. 1). Taken together,
these basins include ~10 tidal channels having similar area (width × length) to the Shibsa
River. We take the tidal flow through these systems to be broadly similar given the linear
relationship between peak tidal discharge and the cross-sectional area of large tidal
channels (Rinaldo et al., 1999), plus the fact that land-surface elevation and tidal range are

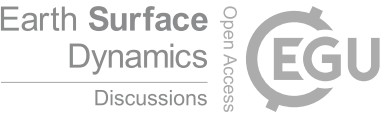

similar across the region (Chatterjee et al., 2013). Thus, even at a first-order, estimates of
total mass transport across the tidal region would well exceed the $\sim 11 \times 10^{11}$ m$^3$ total
volume discharged by the mainstem GBM rivers.
This comparable magnitude of tidal water exchange in the study area and the freshwater
discharge of the GBM rivers demonstrates how tides hold comparable importance in
controlling landscape development in the GBMD, which was suggested as far back as
Galloway (1975). To consider geomorphic importance further, we make analogous
estimations of sediment transport ($Q_s$) that supports land-surface aggradation and the
dominant water discharge ($Q_{dom}$) that controls tidal channel morphology (Rinaldo et al.,
1999). As done for water discharge, by taking the average of our tidal hydrography data for
sediment transport we calculate a mean annual exchange of suspended sediment through
the Shibsa River tidal station to be $\sim 1 \times 10^{11}$ kg ($\sim$100 Mt).  For comparison, this estimate
of sediment load is roughly 15% of the $\sim$700 Mt of sediment annually discharged to the
coast by the GBM rivers (Goodbred and Kuehl, 1999).  Thus, if we extrapolate any similar
transport value to the other nine GBM tidal channels, then the sediment exchange through
the tidal channels is easily found to be comparable to the main river mouth.  There is, of
course, the very important caveat that tidal sediment transport is not unidirectional, and so
this integrated exchange of tidal sediment is not a net flux, as it is for the river sediment
discharge.  Nevertheless, the relevant point is that local, geomorphic reaches of the tidal
delta plain have the opportunity for landscape building through tidal water and sediment
exchange at a similar magnitude to the mainstem GBM rivers.  This assertion is not
surprising given the relative stability of the tidal delta plain, which experiences relatively
little net erosion ($\sim$4 km$^2$/yr, or $\sim$0.02% annual loss; Sarwar and Woodroffe, 2013) and is
offset by widespread sediment deposition on both land-surface (Rogers et al., 2013) and in
channels (Wilson et al., 2017).
From this study, we understand that tidal energy, independent of the main river mouth,
accounts for a twice-daily exchange of a mass equivalent to 4-15% of the yearly averaged
daily GBM river discharge.  In primary channels, the magnitude of this exchange is
controlled more by the spring-neap tidal variability rather than the seasonal input of new
material (Fig. 4). In the smaller BR channel, on the other hand, SSC variability demonstrates
profound seasonality, presumably because discharge (and therefore stream power) is at
least an order of magnitude smaller here than in the Shibsa River. This disparity is
important when we consider land-building processes, as the majority of the SNF is
plumbed by channels on the scale of the BR or smaller.
These findings emphasize the essential role that tides play in maintaining the largest
portion of the GBM lower delta plain, which is not under direct river influence.  However,
despite the essential role of tides in sediment dispersal to large areas of the delta, the
supply of sediment remains almost wholly derived from the river mouth and largely
contemporaneous with seasonal fluvial discharge, especially in the secondary and tertiary
channels that irrigate the SNF.  Together, the coupled system in which the GBM rivers
deliver sediment that is subsequently redistributed by tidal energy is fundamentally
responsible for sustainability of this region relative to sea-level change. A significant



corollary of this fact is that a change in sediment supply from the GBM rivers, such as that
proposed under the India River Linking Project, could pose a serious threat to delta
sustainability (Higgins et al., 2018).
To summarize, as the central coastal region receives little direct water and sediment
discharge from the GBM, the results herein emphasize that tidal exchange is the  dominant
geomorphic agent in the region with a mass and energy exchange of comparable or greater
magnitude to the mainstem rivers. It is, of course, essential to recognize that most
freshwater and sediment exchanged within the tidal system is ultimately sourced by the
main rivers, and that these are intrinsically coupled systems. Thus, continued sustainability
of the region will require the sustained delivery and exchange of water and sediment
between the fluvial and tidal portions of the delta.

**5.2 – Sedimentation in the Sundarbans and Infilling of Tidal Channels**

Our observations of tidal sediment exchange provide a useful baseline for examining
sedimentation in the Sundarbans and broader tidal delta plain, which are at risk from sea-
level rise and inundation without an adequate supply of sediment.  To date, the best
estimate of total sedimentation in the Sundarbans is $1.1 \times 10^{11}$ kg/year (~100 Mt), based
on one season of direct sedimentation measures at 48 stations across the region (Rogers et
al., 2013). This mass of sediment deposited in the Sundarbans is basically equivalent to the
~100 Mt of sediment that we observe transported through the Shibsa River transects.
Thus, recalling that our local measurements likely capture just 5-10% of total suspended
sediment transported through the tidal channels of the region, it becomes evident that
there is generally adequate suspended sediment available to support regional
sedimentation in the Sundarbans.
Another plausible implication is that there appears to be adequate sediment available for
the restoration of land elevation within the poldered region, which is a major challenge
facing coastal Bangladesh (Amir et al., 2013).  Although a definitive answer remains to be
determined, this general assertion is supported by observations of the rapid sedimentation
that occurred on Polder 32 following embankment failures during cyclone Aila in 2009
(Auerbach et al., 2015).  Measurements at Polder 32 after these failures found an average of
37±17 cm/yr of tidal sedimentation sustained over a two-year period, corresponding to a
total annual deposition of ~5 Mt. Based on inundation depth and period, this accounts for
an average of ~0.2 g/L for the water that flooded the island during this time. This value
compares to a mean suspended sediment concentration of ~0.6 g/L measured during our
hydrographic surveys, suggesting that roughly one-third of the tidal sediment inundating
the landscape generated these very rapid sedimentation rates. Ultimately, limitations in the
present data preclude a closed, precise sediment budget, but our collective observations
over several different studies remain consistent in direction and magnitude. These indicate
persistent, relatively rapid, rates of deposition that are sustained by the large-magnitude
conveyance of sediment through the tidal channels and ultimately supplied by seasonal
discharge of the mainstem rivers (Rogers et al., 2013; Auerbach et al., 2015; this study).




Upstream of our transect sites, the landscape is almost entirely embanked by polder
systems. With limited opportunity for sediment deposition on this formerly intertidal
platform, and with the resulting reduction in the tidal prism upstream, channel
sedimentation and infilling has become a major problem. Wilson et al. (2017) demonstrate
that by preventing the inundation of the intertidal platform, poldering has reduced the tidal
prism of the broader southwest region by as much as $1.4 \times 10^9$ m³. If we assume that this
volume reduction is relatively evenly dispersed across the delta plain, then it would have
led to a 25-50% reduction in the local tidal prism measured at our sites. These effects are at
least partially responsible for the ~1400 km of channel infilling that has taken place over
the last few decades, resulting in the creation of new agriculture and aquaculture
opportunities but also altering drainage, transportation routes, and feedback responses of
the regional tidal hydrodynamics (Wilson et al., 2017). The mass of sediment that has
infilled these channels is calculated to be $6.15 \times 10^{11}$ kg, which would be ~$1.2 \times 10^{10}$ kg/yr
assuming a roughly constant rate (Wilson et al., 2017). Of these infilled channels, ~15%
(~200 km) are part of the former channel network connecting upstream of our northern
transect (Fig. 1). Thus, a proportional rate of sedimentation lost to these channels would be
~$0.18 \times 10^{10}$ kg/yr, which is ~25% of the estimated $0.68 \times 10^{10}$ kg fluxing through the
northern transect (to the north) each year.  While this sediment flux is four times greater
than the expected total based on infilling rates from Wilson et al. (2017), it relies on the
same previously described assumptions (i.e., no lateral exchange with neighboring rivers,
non-end-member flux reflecting an average of end-member conditions). More importantly,
it appears that there is sufficient sediment available to continue infilling channels, and
future studies should constrain whether this region is, in fact, infilling faster than other
areas on the tidal delta plain, as this would hold important implications for regional
navigation and hydrodynamic changes.
**6 – Conclusions**
In the present study, we have measured tidal and seasonal variability associated with
water discharge and suspended sediment concentration (SSC), and used these observations
to compute the magnitude of water and sediment exchange through a single tidal channel.
As has been suggested previously, the wet season is found to exert a strong control on the
timing and magnitude of sediment transport in this system, despite seemingly modest
changes to the hydrodynamics. Indeed, despite a reduced tidal range and similar peak SSC,
sediment transport during the monsoon is always of greater magnitude than during the dry
season. Understanding this relationship is critical for planning any potential land recovery
strategies in the future.  The importance of the monsoon also provides a new perspective
into the meaning of a "tidal delta." While it is clearly the tides that perform much of the
work to shape the delta – including driving a net flood-oriented direction of sediment flux –
it is the seasonal influx of riverine sediment that allows this work to continue. Finally, this
research demonstrates that the mass of sediment transported north of our study area is
more than sufficient to fill channels and create additional land. Ideally, future land-use
management strategies should divert some of this excess sediment into polder interiors
through tidal river management, and allow this landscape to continue to prosper.



**Code availability:**
**Data availability:**
Data used for this publication will be archived in the Marine Geoscience Data System.
**Sample availability:**
Samples from this publication are stored in the sedimentology laboratory at Vanderbilt
University
**Author Contribution:**
The experiment was designed by RH and SG, with input from RB and JB. RH and RB lead the
field research efforts with support from SG and JB. RH wrote the majority of the manuscript
and figures, with substantial input from SG. RB and JB also contributed to the manuscript
and figures.
**Competing interests:**
The authors declare that they have no conflict of interest.

**Acknowledgements:**
This work would not be possible without the support of our local collaborators, Drs. Kazi
Matin Ahmed and Syed Humayun Akchter from Dhaka University, who oversee in-country
logistics and offer local guidance. We would also like to thank Abu Naser Hossain of the
Forestry Crime Department for his help with permitting, and Nasrul Islam Bachchu of
Pugmark Tours, and the captain and crew of the M/V Bawali and M/L Mawali for their
seemingly endless patience with our field logistics. We would also like to thank Md.
Saddam Hossain, Abrar Hossain, Mynul Hassan, Carol Wilson, and Mike Reed for their field
support. This research was supported by the US Office of Naval Research (N00014-11-1-
0683) and the National Science Foundation (Coastal SEES- #1600319).



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



Table 1: Measurements of sediment flux and tidal prism from the Shibsa River. Shaded
rows represent measurements taken during spring tides.

| | Transect | Tidal Range (m) | Tidal Prism (m³) | | | Sediment Load (kg) | | |
|---|---|---|---|---|---|---|---|---|
| | | | Ebb | Flood | Net | Ebb | Flood | Net |
| Dry Season | South | 2.1 | 2.00E+08 | -2.00E+08 | 4.30E+05 | 2.05E+07 | -4.70E+07 | -2.66E+07 |
| | North | 2.2 | 1.40E+08 | -1.50E+08 | -1.30E+07 | 1.55E+07 | -2.37E+07 | -8.21E+06 |
| | South | 5.5 | 4.50E+08 | -4.30E+08 | 2.30E+07 | 1.83E+08 | -2.30E+08 | -4.69E+07 |
| | North | 5.7 | 3.10E+08 | -2.30E+08 | 7.90E+07 | 2.15E+08 | -1.90E+08 | 2.49E+07 |
| Monsoon | South | 2.7 | 2.64E+08 | -1.81E+08 | 8.28E+07 | 4.47E+07 | -3.89E+07 | 5.77E+06 |
| | North | 2.2 | 1.83E+08 | -1.06E+08 | 7.69E+07 | 6.20E+07 | -4.12E+07 | 2.08E+07 |
| | South | 4 | 4.71E+08 | -5.12E+08 | -4.16E+07 | 3.20E+08 | -3.85E+08 | -6.50E+07 |
| | North | 3.9 | 2.40E+08 | -2.85E+08 | -4.43E+07 | 2.54E+08 | -3.31E+08 | -7.65E+07 |




**Figures:**

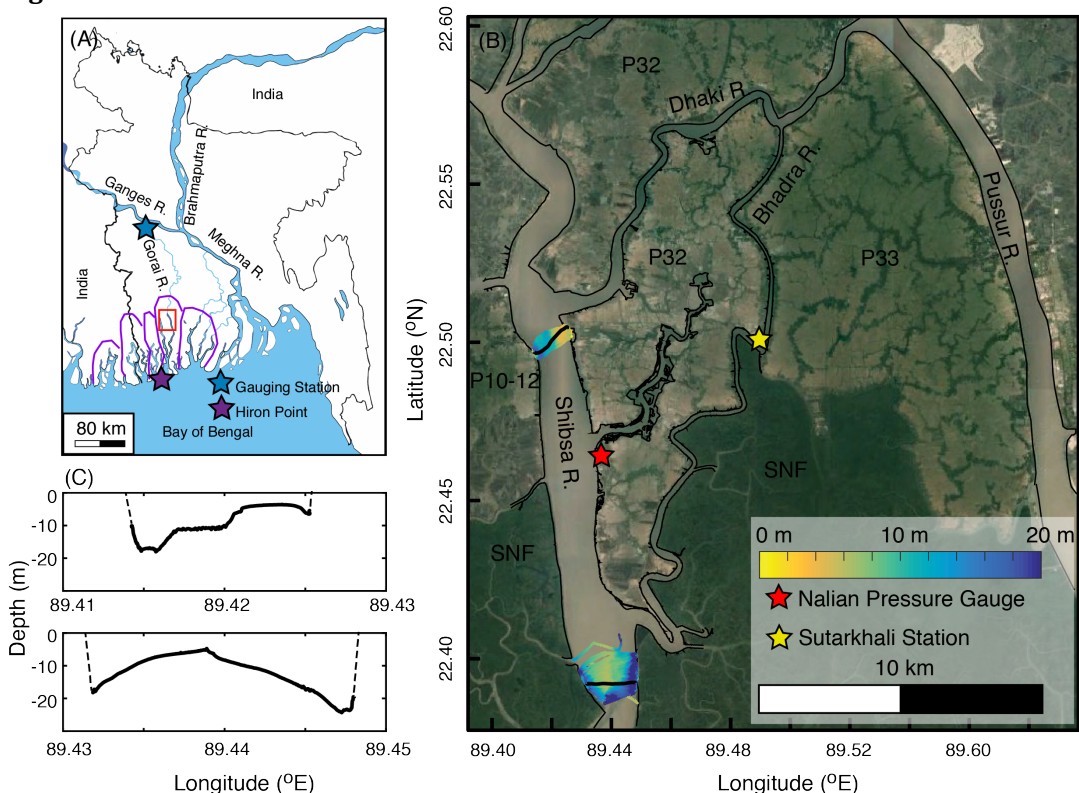

Fig. 1 – A) Location of Bangladesh and the specific region of interest for this study, as well
as the approximate outlines of the five major tidal distributary basins of the SW delta in
purple. B) Satellite image of P-32 study area, with bathymetry overlain in the regions of the
northern and southern transects. Long-term and short-term pressure sensor locations are
also identified. C) Characteristic river cross sections for the northern and southern
transects. The specific transects used for these cross sections are highlighted in black in
(B).

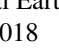



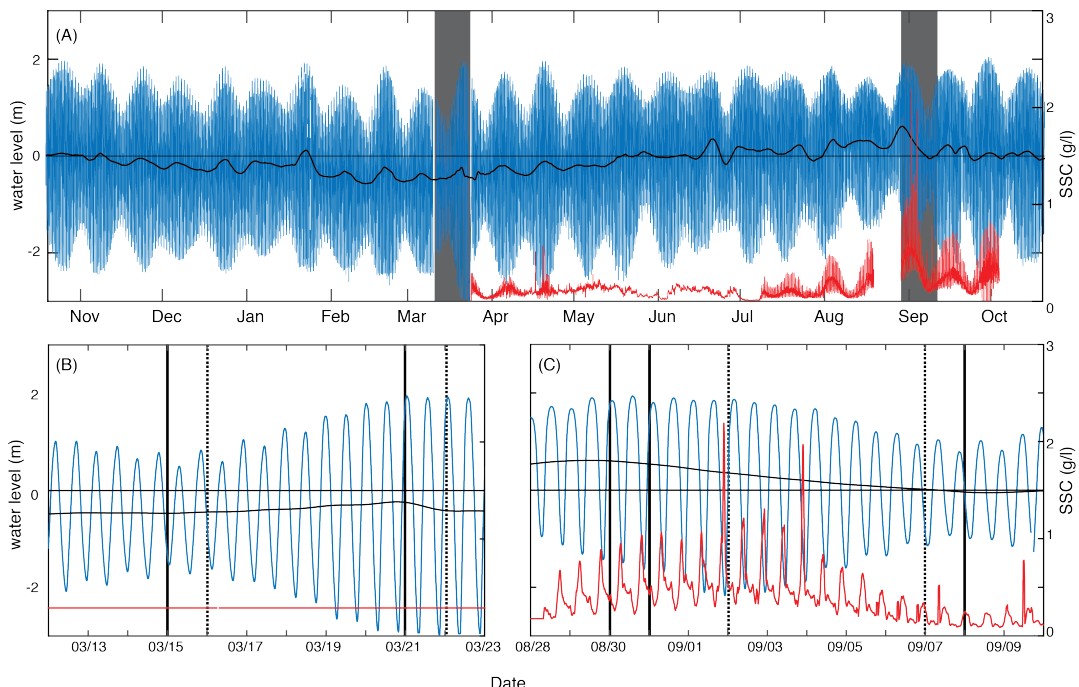

Fig. 2 – A) Long-term water level elevation (blue) and suspended sediment concentration
(red) recorded at Sutarkhali. Black is the tidally filtered water level to highlight seasonal
trends of relatively higher water during the monsoon, despite similar maximum tidal
elevation. Note also the arrival of increased SSC associated with monsoon discharge of the
GBM, beginning in August. Areas shaded in gray depict the periods of focused field work,
highlighted below in panels (B) and (C). Days where transect measurements were recorded
are noted with vertical black lines, where solid are from the southern transect, and dashed
are from the northern transect. In (B), the horizontal red line represents the maximum SSC
observed in the spring-neap tidal cycle following our focused field work, as SSC was not
measured at this location previously.



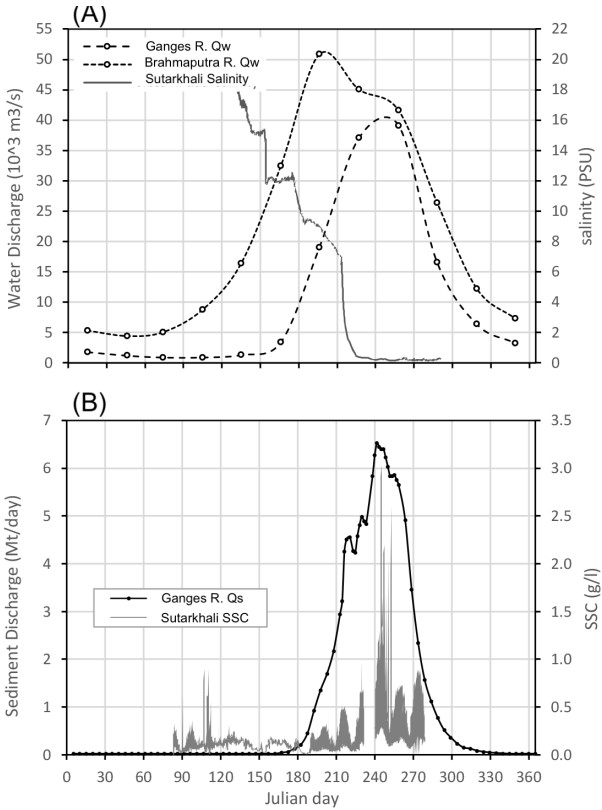

Fig. 3 – A) Ganges and Brahmaputra River water discharge (Qw), and salinity measured at
Sutarkhali Station, demonstrating the reduction in P-32 salinity associated with the arrival
of freshwater from the GBM rivers. B) Ganges river sediment discharge (Qs) and SSC
measured at Sutarkhali station, demonstrating the increase in local SSC coincident with the
peak SSC discharge of the Ganges R.



Earth **Surface**
**Dynamics**
Discussions
EGU

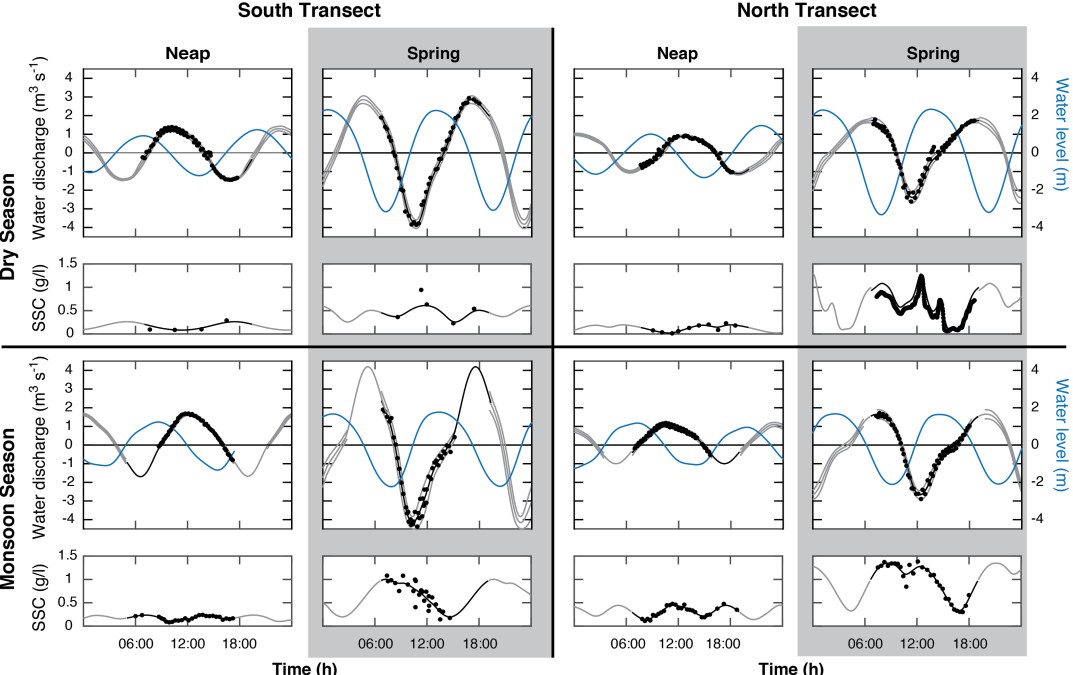

Fig. 4 – Instantaneous water discharge, water level, and depth and width-averaged SSC for
each day of cross-channel transects. Dry season measurements are in the upper half, while
monsoon season transects are on the bottom. Spring tides in either season are shaded in
gray. The two left columns are southern measurements, and the two right columns are
from the northern transect. Black dots correspond to specific measurements, while gray
lines represent the estimated error, tile forwards and backwards by 12.4 hours. For
discharge, dashed lines in the monsoon represent maxima based on extrapolations from
the dry season ratio. While seemingly unreasonable, they are provided here for context.





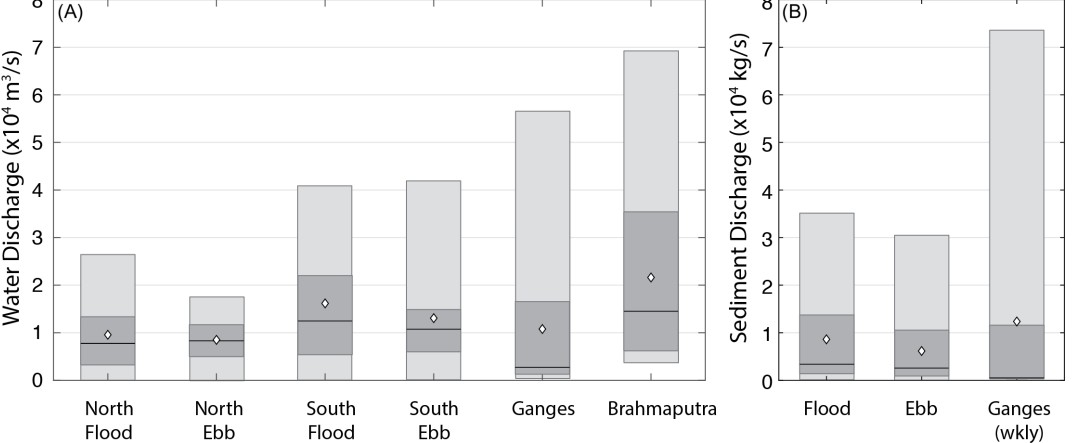

Fig. 5 – Comparison of mean (diamond), median (black line), 25th and 75th percentile (lower and upper limits of darkly shaded box) and total range (lightly shaded box) for water discharge (A), and sediment discharge (B). A) demonstrates that median and mean discharge along either transect are comparable to those of either the Ganges or Brahmaputra River. B) demonstrates that as with water, mean sediment discharge on both the flood and ebb tides is approximately the same as the weekly averaged Ganges sediment discharge.

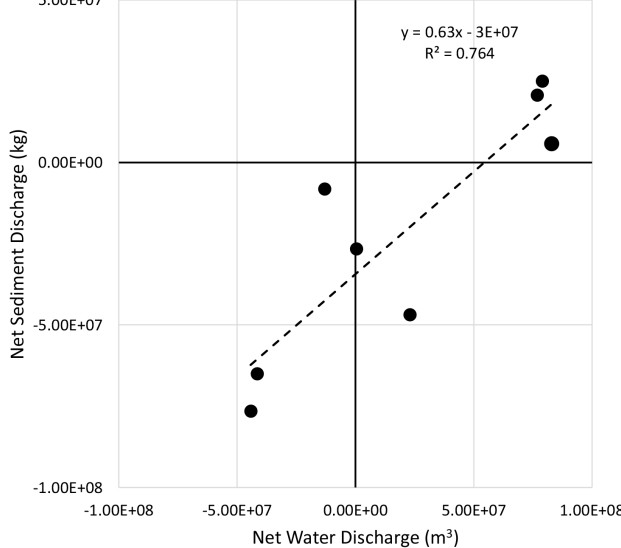

Fig. 6 – Net water discharge vs. net sediment discharge for all of the survey days on the Shibsa River. As expected, we observe a positive trend to this relationship. The negative y-intercept of the best-fit curve demonstrates the overall flood-oriented nature of sediment transport in this tidal channel.