# Peer review of "Observations and scaling of tidal mass transport across the lower Ganges-Brahmaputra delta plain: implications for delta management and sustainability"

_Earth Surface Dynamics, 2018_

## Referee Comment (RC1) · J. Shaw (Referee) · 26 Oct 2018

This study details field observations of tides and sediment transport in the tidal region of the Ganges-Brahmaputra-Mengha Delta system. This research is important because of the dearth of direct measurement in this vast system, and provides first insights about how the delta keeps pace with relative sea level rise, context for recent human-induced changes, and a baseline for proposed large scale water projects. The authors characterize tidal range, tidal prism, and sediment transport at a few key sites on primary and secondary distributary channels in the tidal region of the delta. This

manuscript significantly increases our understanding of this system. I have a few questions, but I think that this paper should be published in ESURF after minor revisions. The sediment concentration and transport data are the most important deliverable to me, but I have a hard time summarizing the findings, because they seem contradictory. Point 1: suspended sediment concentration in a secondary channel increases during the wet season by three fold, indicating a fluvial origin (Figure 2). Point 2: Surveys of net sediment discharge in a primary channel collected over all survey days reveal a net import of sediment (Figure 7), which suggests that sediment transport is primarily dependent on net water discharge, which suggests that freshwater arrival is of secondary importance. However, the flux variation here is also about a factor of three or four (Table 1), consistent with the secondary (BR) channel. I encourage the authors to test the hypothesis that the transport of sediments is really controlled by the same thing in both primary (Shibsa) and secondary (BR) channels. I understand that this is difficult to do given the varying data types, but that is a simpler and more tractable explanation.

Minor comments

L258: I think that this is a relatively weak reason to ignore bedload. My intuition is that lots of bed material sand can become suspended under achievable shear velocities and contribute to SSC measurements during velocity maxima, and be transported onto secondary channels or islands if there is enough water discharge. I would say you can neglect bedload if there are no bedforms in your multibeam surveys. Otherwise, I think you just need to say that it could be happening, but that it's is likely far less than the suspended component and necessarily neglect it from surveys.

L261: I do not know what "tiling observations" means. Perhaps a quick definition is in order.

L339: It took me a minute to figure out that the tidal prisms you are measuring are from integrating the discharge. I imagine prisms as a space filled, which would be impossible to measure. Consider defining how prisms are found.

**ESurfD**
L459: Total _annual_ mass transport

L496: I do not understand how sediment moving through the system could be "almost wholly derived from the river mouth," but that the flux through the fiver major tidal channels could be estimated as roughly equal to the sediment flux of the main river (L486-487). I would suspect that there could also be significant re-entrainment of continental shelf or island sediments that were once river derived, but have been in the coastal zone for years or maybe far longer than that. I think that the case for re-entrained sediments can't be disproved here.

L555: led to a reduction in tidal prism. . . assuming no feedbacks to tidal dynamics, correct?

---

## Referee Comment (RC2) · Allison (Referee) · 27 Nov 2018

The Hale et al. manuscript is a fine addition to the very sparse literature on water and sediment dynamics in the Ganges-Brahmaputra coastal zone. I think the paper, which should be published, could be improved in several ways.

1. The dataset is sparse, which is understandable given the difficult logistical conditions to work in this setting. However, absence in particular of CTD cast data synchronous with the OBS cast data, left a number of questions in my mind about the possibilities of

water column salinity stratification in the channels during the dry season, and sediment stratification and bed storage and or sediment convergence during both studies at slack periods and seasonally. I realize that the authors can't fully address these issues, but I think some of the questions could be allayed by presenting some of the original data– ADCP transects of velocity magnitude, direction and backscatter intensity, and OBS profiles for these example sections. None of this data that is used to calculate fluxes is presented as is, and, seeing some of it would be beneficial to the reader.

2. The methodology is lengthy. If necessary, it could be split off into a supplementary methods section, that would allow greater detail on some of the data manipulations to arrive at fluxes that were only briefly covered in the existing version.

3. I believe some mention of the potential importance of tropical cyclones needs presenting in the intro and discussion. That is, these large events may have an impact on sediment fluxes in the system that may or may not exceed the seasonal and tidal scale processes. Although there is no data presented here, it should be mentioned as a possible and unresolved control in the system.

4. line 166. OBS's do not measure SSC's, they measure turbidity and have to be calibrated. Hence, while the profiler OBS was calibrated as discussed, how did SSC's get derived for the long-term station at Sutarkhali?

5. line 256. This mentions ignoring bedload transport but, what is neglected is sand transport in suspension (bed material load transport). Since water sampling was not done isokinetically (Niskin), this component was missed or undersampled. It appears from the water flux rates (no adcp velocity profiles shown) that the tidal energies are high enough during max ebb and flood to transport sand. I would mention this caveat to be fair about what you are actually measuring (fine flux).
* * *
**ESurfD**

---

## Author Comment (AC1) · 21 Dec 2018

This study details field observations of tides and sediment transport in the tidal region of the Ganges-Brahmaputra-Mengha Delta system. This research is important because of the dearth of direct measurement in this vast system, and provides first insights about how the delta keeps pace with relative sea level rise, context for recent human-induced changes, and a baseline for proposed large scale water projects. The authors characterize tidal range, tidal prism, and sediment transport at a few key sites on primary and secondary distributary channels in the tidal region of the delta. This

manuscript significantly increases our understanding of this system. I have a few questions, but I think that this paper should be published in ESURF after minor revisions. The sediment concentration and transport data are the most important deliverable to me, but I have a hard time summarizing the findings, because they seem contradictory. Point 1: suspended sediment concentration in a secondary channel increases during the wet season by three fold, indicating a fluvial origin (Figure 2). Point 2: Surveys of net sediment discharge in a primary channel collected over all survey days reveal a net import of sediment (Figure 7), which suggests that sediment transport is primarily dependent on net water discharge, which suggests that freshwater arrival is of secondary importance. However, the flux variation here is also about a factor of three or four (Table 1), consistent with the secondary (BR) channel. I encourage the authors to test the hypothesis that the transport of sediments is really controlled by the same thing in both primary (Shibsa) and secondary (BR) channels. I understand that this is difficult to do given the varying data types, but that is a simpler and more tractable explanation.

Minor comments

L258: I think that this is a relatively weak reason to ignore bedload. My intuition is that lots of bed material sand can become suspended under achievable shear velocities and contribute to SSC measurements during velocity maxima, and be transported onto secondary channels or islands if there is enough water discharge. I would say you can neglect bedload if there are no bedforms in your multibeam surveys. Otherwise, I think you just need to say that it could be happening, but that it's is likely far less than the suspended component and necessarily neglect it from surveys.

L261: I do not know what "tiling observations" means. Perhaps a quick definition is in order.

L339: It took me a minute to figure out that the tidal prisms you are measuring are from integrating the discharge. I imagine prisms as a space filled, which would be impossible to measure. Consider defining how prisms are found.

**ESurfD**
[Figure]

L459: Total _annual_ mass transport

L496: I do not understand how sediment moving through the system could be "almost wholly derived from the river mouth," but that the flux through the fiver major tidal channels could be estimated as roughly equal to the sediment flux of the main river (L486-487). I would suspect that there could also be significant re-entrainment of continental shelf or island sediments that were once river derived, but have been in the coastal zone for years or maybe far longer than that. I think that the case for re-entrained sediments can't be disproved here.

L555: led to a reduction in tidal prism... assuming no feedbacks to tidal dynamics, correct?

[Figure]

[Figure]
The Hale et al. manuscript is a fine addition to the very sparse literature on water and sediment dynamics in the Ganges-Brahmaputra coastal zone. I think the paper, which should be published, could be improved in several ways.

1. The dataset is sparse, which is understandable given the difficult logistical conditions to work in this setting. However, absence in particular of CTD cast data synchronous with the OBS cast data, left a number of questions in my mind about the possibilities of

water column salinity stratification in the channels during the dry season, and sediment stratification and bed storage and or sediment convergence during both studies at slack periods and seasonally. I realize that the authors can't fully address these issues, but I think some of the questions could be allayed by presenting some of the original data– ADCP transects of velocity magnitude, direction and backscatter intensity, and OBS profiles for these example sections. None of this data that is used to calculate fluxes is presented as is, and, seeing some of it would be beneficial to the reader.

2. The methodology is lengthy. If necessary, it could be split off into a supplementary methods section, that would allow greater detail on some of the data manipulations to arrive at fluxes that were only briefly covered in the existing version.

3. I believe some mention of the potential importance of tropical cyclones needs presenting in the intro and discussion. That is, these large events may have an impact on sediment fluxes in the system that may or may not exceed the seasonal and tidal scale processes. Although there is no data presented here, it should be mentioned as a possible and unresolved control in the system.

4. line 166. OBS's do not measure SSC's, they measure turbidity and have to be calibrated. Hence, while the profiler OBS was calibrated as discussed, how did SSC's get derived for the long-term station at Sutarkhali?

5. line 256. This mentions ignoring bedload transport but, what is neglected is sand transport in suspension (bed material load transport). Since water sampling was not done isokinetically (Niskin), this component was missed or undersampled. It appears from the water flux rates (no adcp velocity profiles shown) that the tidal energies are high enough during max ebb and flood to transport sand. I would mention this caveat to be fair about what you are actually measuring (fine flux).

[Figure]

In Response to R1:

Dear Dr. Shaw, I think your assessment of our major findings is accurate, except that I see no contradiction. In the primary channel, we maintain that tidal stage (i.e., discharge) is the dominant control on suspended sediment concentrations year round, with the introduction of GBM-derived material during the monsoon playing a secondary role. In the smaller channels (e.g., Bhadra), seasonal delivery of new material appears to play a larger role in sediment flux. This introduces the idea of a "discharge threshold" above which seasonal sediment delivery is relatively less important, however future research is required to test this idea.

L258 – Point taken. This sentence was introduced to let the reader know that we are not forgetting about bedload. The revised manuscript will be more explicit.

L261 – Text will be updated to explain that "tiling observations" means repeating the measured time series at 12.4-h increments to improve interpolation/extrapolation accuracy.

L339 – We will make sure to define tidal prisms as the integrated discharge in this context.

L459 – This change will be made.

L496 –We don't mean to imply that extensive mixing is not happening – quite the opposite. Several studies (e.g., Rogers et al., 2013; Rogers and Overeem, 2017) have demonstrated the presence of a weak, excess-[7]Be signal in sediment accumulating on the mangrove platform during the monsoon season, with mixing/dilution offered as one potential explanation. This comparison was intended to provide the reader with a sense of scale. We have reworded to be more clear about our intended meaning.

L555 – In this case, the "volume reduction" refers to poldered area that would have been flooded by the Shibsa and its distributaries, multiplied by a characteristic flooding depth. We have reworded this as "reduction or redistribution" to clarify.

In response to R2:
Dear Dr. Allison, Thank you for your review if our manuscript. Please consider the following responses to your concerns:

- As you mention, field logistics are challenging here, and for most of this research we did not have a profiling CTD at our avail. Anecdotally, we observe sediment laden plumes regularly boiling to the surface during energetic tides in all seasons, suggesting a physically well-mixed system. Shaha and Cho (2016) demonstrate minimal stratification in primary channel (Pussur) regardless of season, although they do indicate that early in the wet season (e.g., July) mixing between Shibsa and Pussur channels (which occurs in the Dhaki) can result in vertical stratification. Wilson et al. (AGU conference 2018) published observations of surface conductivity along a transect extending from the study area to the Bay of Bengal coastline in March 2015 (dry season). They demonstrate a consistent increase from P-32 (~24 mS/cm) to the coast of the Bay of Bengal (~40 mS/cm), supporting again that water columns are vertically mixed, rather than stratified.

We have included a figure with example ADCP cross-section velocity data, and SSC casts, to further transparency into our method.

– We will have added more detail to this calculation in the final version.

– Good point. We are well aware of the regional importance of tropical cyclones, and their potential to move sediment. We will update the text to include this discussion.

– Thank you for this clarification. In fact, the OBS deployed in the tidal channel was the same instrument used in the dry season field work (again – we were instrument-limited). The calibration was built from the >100 filtered water samples. We have restructured the methods to describe this calibration earlier on.

– Thank you. A similar concern was raised by Referee 1, and we invite you to consider our response to them.

[revised manuscript text omitted]